# Influence of Blind Spot Assistance Systems in Heavy Commercial Vehicles on Accident Reconstruction

**DOI:** 10.3390/s24051517

**Published:** 2024-02-26

**Authors:** Thomas König, Daniel Paula, Stefan Quaschner, Hans-Georg Schweiger

**Affiliations:** CARISSMA Institute of Electric, Connected and Secure Mobility (C-ECOS), Technische Hochschule Ingolstadt, Esplanade 10, 85049 Ingolstadt, Germany; daniel.paula@thi.de (D.P.); hans-georg.schweiger@thi.de (H.-G.S.)

**Keywords:** accident analysis, accident reconstruction, road safety, vehicle active safety

## Abstract

Accidents between right-turning commercial vehicles and crossing vulnerable road users (VRUs) in urban environments often lead to serious or fatal injuries and therefore play a significant role in forensic accident analysis. To reduce the risk of accidents, blind spot assistance systems have been installed in commercial vehicles for several years, among other things, to detect VRUs and warn the driver in time. However, since such systems cannot reliably prevent all turning accidents, an investigation by experts must clarify how the accident occurred and to what extent the blind spot assistance system influenced the course of the accident. The occurrence of the acoustic warning message can be defined as an objective reaction prompt for the driver, so that the blind spot assistance system can significantly influence the avoidability assessment. In order to be able to integrate the system into forensic accident analysis, a precise knowledge of how the system works and its limitations is required. For this purpose, tests with different systems and accident constellations were conducted and evaluated. It was found that the type of sensor used for the assistance systems has a great influence on the system’s performance. The lateral distance between the right side of the commercial vehicle and the VRU, as well as obstacles between them, along with the speed difference can have great influence on the reliability of the assistance system. Depending on the concrete time of the system’s warning signal, the accident can be avoided or not by the driver when reacting to this signal.

## 1. Introduction

Accidents involving pedestrians and cyclists, known as vulnerable road users (VRUs) by the international convention, are disproportionately likely to result in serious or fatal injuries and play an important role in accident reconstruction. One possible scenario in the inner-city environment is an accident between right-turning commercial vehicles and parallel pedestrian or bicycle traffic [1,2]. Due to the elevated driver’s seat position and the non-full-surface glazing of the driver’s cabs, areas near the commercial vehicle cannot be easily seen by the driver. These areas are called blind spots. To minimize this problem, the EU Directive 71/127/EEC [3] has prescribed several complementary mirror systems for commercial vehicles since 1971. As an additional problem, a heavy commercial vehicle’s driver must pay attention to various potentially critical areas simultaneously, especially when turning in a limited space in an inner-city environment. It is often impossible to concentrate fully on a specific potential danger zone during the entire turning process. Cyclists move at a comparatively high speed and may only be perceptible for a short time in one of the exterior mirrors and can therefore be overlooked by the commercial vehicle’s driver [2,4,5,6,7].

To compensate for this problem and thus noticeably reduce the number of traffic accidents, Regulation EU 2019/2144 of 27.11.2019 [8] successively introduced various driver assistance systems as mandatory equipment for newly type-approved or newly registered vehicles. In particular, to improve the perceptibility of VRUs for commercial vehicles with a maximum permissible mass of more than 3.5 t, a so-called blind spot assistant is required from 6 July 2022 onwards for newly type-approved vehicles and from 7 July 2024 for newly registered vehicles. Due to the increased risk potential because of their high speed, cyclists are explicitly targeted. However, the functionality of the blind spot assistance system also includes comparable road users, such as users of small electric vehicles and pedestrians. The technical characteristics of this assistance system and the required test methods are described in UN-ECE R151 of 25 September 2020 [9] and summarized in the following paragraph. 

Two escalation levels are defined here: The information signal is intended to alert the commercial vehicle driver to a bicycle riding on the right-hand side of the vehicle in the potential danger zone when a turning maneuver is initiated. It is said to be an optical signal appearing in the right-hand area of the vehicle cabin. The warning signal must be activated when the system detects an imminent collision with a bicycle in the danger zone if the driver does not react using an appropriate steering angle or an operation of the direction indicator. This can be an optical, acoustic, haptic signal, or a combination of them. While this optical–acoustic warning signal may be switched off manually or automatically, the visual information signal may only be deactivated automatically in case of a system malfunction or contamination of the sensors. 

The assistance system must work in a self-speed range between 0 km/h and 30 km/h. On the one hand, it is intended to warn of a cyclist crossing in front of the vehicle in an area that is not directly visible when the vehicle is stationary. On the other hand, a warning signal should be issued if a bicycle moves at a speed between 5 km/h and 20 km/h on the right side of the vehicle. The detection area should be in a lateral corridor between 0.9 m and 4.25 m to the right of the vehicle, which extends to 7.00 m in front of and 30.00 m behind the right front corner. In addition, it should be possible to perceive a bicycle at a lateral distance of 0.25 m to 0.90 m at the level of the foremost front wheel and to issue an information signal [9]. Speeds in the unit km/h are given with an accuracy of 1 km/h, and two valid decimal places are given for distances in the unit m.

UN-ECE R151 defines two static and one dynamic test method, whereby only the timely occurrence of the information signal is assessed as a passing criterion. The warning signal, which indicates an imminent collision, is not considered [9]. In the static test, the vehicle is at a standstill, while in the first test run, a bicycle approaches from the right at a speed of 5 km/h and a distance of 1.15 m in front of the vehicle. This must be detected and signaled by the assistance system at least 2.00 m before passing the right-hand front corner of the vehicle. In the second test procedure, the bicycle moves parallel from the rear to the front of the vehicle at a 20 km/h speed at 2.75 m from the right side. In this case, the assistance system must warn the driver at the latest when the bicycle is still 7.77 m behind the front of the vehicle. In the dynamic system tests, the vehicle passes through a corridor while a bicycle moves parallel to it on the right at a lateral distance of 1.25 m or 4.25 m. In addition to the variable lateral distance, alternating speeds of 10 km/h and 20 km/h are also specified for both the test vehicle and the bicycle. This results in different relative speed constellations between the vehicle and the detection object, which depict different traffic scenarios. The test is deemed to have been passed when the blind spot assistance system detects the bicycle at the latest in a certain constellation that depends on the driving speeds and warns the rider. Furthermore, no warning message may be induced when passing a traffic sign set up at the beginning of the driving corridor or when passing a stationary bicycle. 

In addition to this EU regulation, the German Federal Ministry for Digital and Transport (Bundesministerium für Digitales und Verkehr BMDV) launched a funding program in Germany in 2019 for the voluntary retrofitting heavy commercial vehicles with a blind spot assistance system to accelerate the spread of these systems among existing vehicles. In order for a retrofit assistance system to receive a general operating permit from the German Federal Motor Transport Authority (Kraftfahrbundesamt KBA) and to be classified as eligible for funding in this program, it should meet the recommended test criteria published at the national level in the official section of the German state traffic journal (Verkehrsblatt) 19/2018 [1,10]. Here, a lateral coverage area of the blind spot assistance system with a length of 6 m from the front edge of the vehicle and a width of 2.5 m from a lateral distance of 0.9 m to the vehicle is required [11]. Furthermore, an amendment to § 9 of the German traffic regulations (Straßenverkehrsordnung StVO) stipulates that a driver of a motor vehicle with a maximum permissible mass of more than 3.5 t in urban areas must drive at walking speed when turning right if there is a bicycle traffic driving straight ahead on or next to the carriageway or in the immediate area of turning with a cyclist crossing the carriageway pedestrian traffic. The walking speed is not clearly defined in the StVO [12], and there is also no binding decision from the German Federal Court of Justice which the subordinate courts could use as a guide. The range set by the General German Automobile Association (Allgemeiner Deutscher Automobil-Club (ADAC)) and the German district courts and higher courts ranges from about 7 km/h to less than 15 km/h [13,14].

If, despite the measures listed, a turning accident occurs, its course of events often must be clarified by forensic accident analysts [15]. Of particular interest is often the clarification of the circumstances under which the driver of the road transport vehicle could have caused the accident. Since the blind spot assistance system is a pure warning system that does not actively interfere with the vehicle’s driving dynamics and is therefore intended to brake it independently, it must be analyzed whether or when the driver was visually or acoustically alerted to the VRU by the system while approaching the collision site in order to direct his gaze into the relevant mirror and to increase the urgency for braking action.

For this purpose, however, the accident analyst must have access to the connecting facts as to when and how the system warned the driver. UN-ECE R160 [16], in its current form, prescribes the event data recorder (EDR) for recording accident-related data only for vehicles up to a maximum permissible mass of 3.5 t. According to the current UN-ECE R160, the data set does not contain any references to the activities of Advanced Driver Assistance Systems (ADASs) [16]. Accordingly, in commercial vehicles, in particular, there are no digital traces of ADAS activities available that would be stored in the event of a collision and could be read by independent third parties. Suppose the possible influence of ADASs and autonomous driving functions of the commercial vehicle involved in the accident is to be included in the technical assessment. In that case, this can only be performed through tests. The legal perspective on the influence of driver assistance systems and autonomous driving functions on an accident has yet to be clarified. In January 2023, Working Group III—AI Liability in Road Traffic/Liability in Autonomous Driving of the 61st German Traffic Court Conference in Goslar drew up the following recommendation: Until further notice, strict liability towards the injured party in the accident should remain fully with the vehicle owner’s liability insurance in the first step. Since it can nevertheless be assumed that in the future, more system errors will lead to accidents instead of human driving errors, product liability is intended to enable liability insurance companies to make recourse claims against the vehicle manufacturer in the second step [17,18].

Such product liability claims are only possible if a technical expert opinion can prove a significant influence of the ADAS interventions on the course of the accident. Forensic experts must have the necessary connecting facts, as neither digital traces nor basic data from experiments are available. Previous experimental investigations, particularly on blind spot assistance systems, make only qualitative statements on fulfilling national and European test criteria and can only be used to a limited extent for forensic accident analysis [1,10,19].

Throughout this article, numerous experiments were carried out with a wide variety of constellations to identify a relevant parameter set for the pre-crash driving behavior and avoidability analysis during accident reconstruction. From this, a method is derived as to how the turning assistance system should be integrated into the pre-crash and avoidability simulations in the future.

In 2019 and 2021, the ADAC took the abovementioned national funding program of the Federal Ministry for Digital and Transport as an opportunity to qualitatively review the performance and functionality of various retrofit assistance systems available on the market in the sense of a consumer protection test by the applicable regulations. The tested systems differ considerably in terms of the environment sensors used, the activation strategy, and the warning strategy [1,10,19]. Selected systems were subjected to static and dynamic tests by the ADAC by the BMVI recommendations and UN-ECE R 151—in some cases, in a slightly modified or simplified form.

As a core result of the ADAC tests, it should be noted that in the case of an inner-city right-turn at walking speed by the road traffic regulations and with the direction indicator activated at an early stage, all tested blind spot assistance systems can develop their full performance by the standard specifications. Under the ideal conditions depicted in the regulations, the assistance systems thus have the potential to avoid accidents between right-turning commercial vehicles and VRUs by warning the driver. This means that the unavoidable inner-city freight traffic can continue to be handled by heavy commercial vehicles in a fundamentally unchanged form with a considerable increase in safety. The assistance systems contribute to increasing road safety and reducing the number of traffic fatalities without having to take costly technical, infrastructural, or legal measures. Nevertheless, shortcomings of the assistance systems were found in the ADAC test series, for example, it is not to be expected that turning accidents in real traffic can be avoided; thus, a closer look from an accident analysis point of view is expedient [1,10,19]. It must therefore be possible to answer the question as to why the driver could not prevent a particular accident despite using a blind spot assistance system.

## 2. Materials and Methods

In order to be able to investigate the extent to which blind spot control systems can make a real contribution to the prevention of right-turn accidents, test series were carried out. Another aim was to explore how such an assistance system can be integrated into the reconstruction of the accident sequence and the avoidability analysis. First, the relevant influencing variables on the behavior of the blind spot assistance systems were identified. Secondly, data should be generated on the timing of the warnings before a collision and on the relative constellation of the vehicle and the vulnerable road user at the time of warning. To ensure comparability and plausibility of the qualitative results, the test setups were very closely based on the ADAC test series based on the specifications of UN-ECE R151 [1,10,19].

### 2.1. Experimental Design

Two static and two dynamic test setups were used, whereby this designation refers to the commercial vehicle since the VRUs were in constantly motion. In the static tests, the commercial vehicle was turned on with automatic transmission in “drive” mode. The vehicle maintained its position by the driver using the service brake. The parking brake was not used to ensure that the ADAS was working.

The test setup for the first static test with a VRU crossing in front of the vehicle was taken from UN-ECE R151, as shown in Figure 1 below [9] and supplemented by further test speeds. The figure particularly explains the distances between the VRU’s driving line and the front of the vehicle and the start distance of the VRU.

The tests were conducted at VRU speeds of 5 km/h, 10 km/h, 15 km/h, and 20 km/h as described in Table 1. The speed measurement was carried out by a speedometer mounted on the bicycle, which determines the driving speed from the wheel rotation speed to an accuracy of 1 km/h. In each case, the latest possible point in time—the so-called last information point (LIP)—was determined in a speed-adjusted manner at which the blind spot assistance system would have to point out the approaching VRU so that the commercial vehicle driver could safely avoid a collision due to a braking reaction. The LIP is given as a distance value measured between the VRU and the right front corner of the vehicle.

In the second static test described in Figure 2, according to UN-ECE R151 [9], a VRU moved parallel to the longitudinal axis of the commercial vehicle from the rear to the front along its right-hand side. In contrast to UN-ECE R151, two lateral distances of 1.25 m and 4.25 m (d_LAT_ in Figure 2) were investigated in the tests. The VRU started about 44 m behind the front of the commercial vehicle to make sure that the starting point was out of the detection range of the sensor systems.

As in the first test setting, different speeds of the VRU were used as shown in Table 2. The distances were determined with a tape measure SYMRON-S SYS20mt from Tajima Tolls Glückstadt, Germany with an accuracy of 0.01 m and DAkkS-certificated calibration.

Furthermore, the tests were carried out with a lateral distance of 4.25 m in an additional passage with a visual obstruction produced by parked vehicles between the commercial vehicle and the VRU (shown as an example in Figure 3, tests 2.11 to 2.15 in Table 2). To guarantee that the worst-case scenario is tested, we ensured that the cars were parked right in front of the sensors on the side of the commercial vehicle. The gaps between the cars were made as small as possible. The lateral distance between the right side of the commercial vehicle and the parked cars was about 1 m and about 1.5 m between the left side of the cars and the VRU.

Similarly, and following the ADAC’s investigations [1,10,19], dynamic tests were carried out in which the commercial vehicle and the VRU moved parallel to each other at different speed constellations with lateral distances of 1.25 m and 4.25 m, respectively, as described in Figure 4 and Table 3.

First, an equal driving speed of 10 km/h and 20 km/h for both vehicles, respectively, was investigated. Here, both vehicles started together and the experiment began when both reached the test speed. Furthermore, reciprocal overtaking maneuvers were depicted, in which one road user with a speed of 20 km/h overtook the other one moving with a constant speed of 10 km/h.

In the dynamic tests described above, based on UN-ECE R151, no intersection of the commercial vehicle and VRU trajectories, and thus no immediate risk of collision, was provoked. Accordingly, as intended, a maximum of the information signal from the turning assistance system can occur. For this reason, a further test setup has already been designed by the ADAC, which can also be used to provoke the occurrence of the warning signal of the blind spot assistance system. From an initially parallel movement of the commercial vehicle and the VRU with a lateral distance of 1.25 m, the commercial vehicle turns right at a defined point as shown in Figure 5. The VRU is represented by a bicycle dummy pulled by a test vehicle at a defined speed, as a collision between the two road users can occur without timely intervention by the commercial vehicle driver [1,10,19].

As an additional test variable, the right direction indicator of the commercial vehicle was activated in half of the test runs to find out if there is an influence on the sensitivity of the assistance system as described in Table 4.

### 2.2. Test Vehicles and Blind Spot Assistance Systems

The blind spot assistance systems currently available on the market use ultrasonic or radar sensors or camera-based object recognition algorithms for environment detection. Ultrasonic and radar sensors are based on the physical principle of time-of-flight measurement. Sound waves or electromagnetic waves are emitted and reflected by surrounding objects. The distance between the sensor and the object can be calculated from the known speed of sound or light and the elapsed time between emitting and receiving the wave signal.

Ultrasonic sensors can reliably detect objects over a distance of a few meters, almost regardless of their surface properties. Several sensors are required over the entire right side of the vehicle to cover the necessary detection range on a heavy commercial vehicle [20,21]. With the help of ultrasonic sensors, it is impossible to classify static and dynamic objects on the right-hand side of the road.

One advantage of radar technology over ultrasonic sensors is the significantly longer detection range. However, the decisive factor for use in a blind spot assistant is the ability to determine the speed of a detected object and thus classify it into static objects and moving road users. As a result, false positive events due to traffic signs or other static obstacles at the edge of the road can be better avoided with radar sensors [20,21]. Depending on the mounting height of the sensors, which is usually at the height of the vehicle’s main frame below the body, the fields of view of both ultrasonic and radar sensors are limited by static obstacles such as parked vehicles [20,21]. The test vehicle used here was a Mercedes-Benz Actros 2545 (vehicle identification number W1T96302010442521) with a first registration date in July 2020 with an engine output of 330 kW and a gross vehicle weight of 26,000 kg with an S1R blind spot assistance system developed by Mercedes itself. Instead of conventional exterior mirror systems, the test vehicle was equipped with wide-angle cameras as a mirror replacement system, whose fields of view are displayed on screens inside the A-pillars.

In camera-based systems, a downstream software function identifies and classifies objects in the camera’s field of view based on patterns and motion profiles stored in a database. The camera’s image quality depends on the prevailing lighting conditions and weather conditions in a similar way to the human eye. This is where complementary infrared sensors can help. Atypical objects or movement patterns may not be detected or incorrectly classified by the evaluation software. The advantage is the usually high mounting position of the camera on the upper edge of the driver’s cab, which provides a view over obstacles next to the vehicle [20,21]. For the tests, the ICA Turn-Assist AAS blind spot assistance system from the manufacturer AXION AG (Weißenhorn, Germany) was used, which was mounted on a MAN TGM 18.290 with its first registration in September 2021 (vehicle identification number WMAN38ZZ2MY418969) with an engine power of 213 kW and a gross vehicle weight of 18,000 kg. A supplementary camera monitor system (CMS) allows the driver to check the plausibility of warnings from Blind Spot Assist and to identify false positives as such [20,21]. Such a CMS from the manufacturer AXION AG was also attached to the MAN test vehicle.

### 2.3. Experimental Area

The tests were conducted on the asphalted 60 m by 70 m outdoor test area of the CARISSMA Research Center of the Technical University of Ingolstadt, which is described in detail in [22]. Permanently installed and mobile pulley systems are available on the site, which allow for the movement of VRU dummies synchronized with the driving speed of the test vehicle via rope pull systems. The road surface was wet, but during the experiments, there was no rain. The sky was cloudy most of time, so there was no relevant impact of the position of the sun.

### 2.4. Test Equipment

The experiments were recorded from several perspectives with video cameras. For this purpose, several compact action cameras of the type HERO8 Black from the manufacturer GoPro Inc. from San Mateo (CA, USA) were mounted on the driver’s cab of the respective freight vehicle, which filmed the driver’s perspective to the front, to the instrument cluster (Figure 6 right), and to the exterior mirrors on the right A-pillar (Figure 6 left). Another GoPro HERO8 Black was attached to the outside of the cab to record the movement of the VRU parallel to the road vehicle. In addition, the test sequence was filmed using two XA30 video cameras mounted on tripods from the manufacturer CANON from Tokyo (Japan) from the perspectives shown in the test sketches above. Action cameras were placed in front of the instrument display and the a-post inside the truck’s cabin, and outside at the right side of the cabin to film the VRU. The positions of the video cameras were chosen to film the movement of the VRU from the front and in a 90° angle from the side.

To synchronize the videos, a horn signal was played at the beginning of each experiment, the characteristic signature of which could be identified in the audio tracks of each video. The synchronization and post-processing of the videos were carried out with the open-access software Shotcut version 22.06.25 from the manufacturer Meltytech LLC (Oceanside, CA, USA).

Furthermore, with the help of a 2D data logger from 2D-Debus & Diebold Messsysteme GmbH from Karlsruhe, Germany, the position, speed, acceleration, and deceleration behaviors of the commercial vehicle were recorded with a measurement accuracy of +/−2%. The device was mounted on the dashboard and calibrated. The measurement data exported in CSV format were evaluated with the software “WinARace” version 2015.12.1.1 belonging to the measuring device.

As a VRU for the experimental setups 1, 2, and 3, a real person was used as a pedestrian or riding a black mountain e-bike from Rockrider. The person had a height of about 1.75 m and was wearing dark clothes under a yellow warning vest. For experiment number 4, we used an EuroNCAP bicyclist target dummy from 4activeSystems GmbH from Traboch, Austria, which was attached to a full-electric smart fortwo by a steel cable.

As a visual obstacle for test number 2, we used some available passenger cars with a height of at least 1.5 m. For the experiments, we used a Mazda 3, a Skoda Octavia, and a Seat Ibiza.

The simulation in Section 3.5 was carried out with the reconstruction software PC-Crash, version 14.0 of DSD Dr. Steffan Datentechnik GmbH from Linz, Austria.

## 3. Results

### 3.1. Static Tests with Crossing VRU

It was found that the radar-based blind spot assistance system of the Mercedes test vehicle was able to reliably detect the VRU approaching ahead of the truck’s front from the right, regardless of the activity status of the right direction indicator. The results are shown in Table 5. Up to a VRU approach speed of 5 km/h, the corresponding information signal was issued to the driver in good distance before the LIP. This corresponds to the test criterion of UN-ECE R151. At higher VRU speeds, the information signal was only issued after the LIP, so that a timely avoidance reaction by the driver was no longer ensured. Only in one case the system could not detect the VRU at all.

The detection range of the camera-based blind spot assistance system of the MAN test vehicle does not cover the area in front of the truck, so a VRU crossing in front of the truck cannot be detected with this system. Although the area to the right of the truck is also visible to the driver for several meters in front of the truck via the camera monitor system, there is no automated image evaluation here. Accordingly, no information or warning signal was triggered by the VRU crossing in front of the truck in any of the test runs.

From an accident analysis point of view, in the case of an accident between a truck and a VRU crossing in front of it, the following findings can be noted with regard to the blind spot assistance system: A camera-based assistance system can be neglected in this scenario due to the limited field of vision. Radar-based systems can only reliably detect pedestrians in time due to the limited detection range on the side. In the case of VRUs with higher internal speeds, such as cyclists, the detection may be too late for the driver to be able to safely avoid the accident by reacting to the warning signal. The activity status of the right-hand direction indicator does not affect the response of the warning message.

### 3.2. Static Tests with Parallel Moving VRU

The radar system of the Mercedes test vehicle was able to detect a VRU approaching from behind, parallel to the longitudinal axis of the truck, at a lateral distance of 1.25 m to 4.25 m in advance of the LIP, irrespective of the actuation state of the right-hand direction indicator, so that an information signal was issued to the driver. The concrete results are shown in Table 6. If there were parked vehicles between the radar sensors mounted on the side of the test vehicle and the VRU’s line of movement as an obstacle to visibility, the VRU could not be detected by the assistance system at any time.

The camera-based blind spot assistance system of the MAN test vehicle reliably detected the VRU approaching from behind at a lateral distance of 1.25 m irrespective of the actuation of the right-hand direction indicator and was able to issue an information signal to the driver in good time before the LIP, up to a test speed of 10 km/h. The detailed test results are shown in Table 7. At higher test speeds, the information signal was only issued after the LIP and thus too late for the driver to react in time. At higher test speeds, the information signal did not occur until after the LIP and thus too late for the driver to react in time. At a lateral distance of 4.25 m between the movement line of the VRU and the right side of the truck, the VRU could not be detected by the blind spot assistance system of the MAN test vehicle. Upon close examination, it was found that the camera image displayed in the on-cab monitor covered a very large area to the right of the vehicle, and the VRU was visible at an early stage at a distance of 4.25 m. However, the VRU was not detected by the blind spot assistance system because the area actively monitored by the blind spot assistance system only has a width of up to 2.2 m maximum. At greater lateral distances, therefore, the only function of the camera–monitor system is to extend the indirect field of vision of the exterior mirrors. Consequently, the VRU could not be detected when there was an obstacle between the VRU motion line and the truck.

From an accident analysist’s point of view, the following results can be kept in mind: None of the blind spot assistance systems examined can detect a VRU if there is an obstacle between it and the truck. The detection range of radar-based systems is sufficient to detect the truck reliably and in good time, even if there is a lateral distance of up to about 4 m between the truck and the VRU. Due to the limited side coverage area, camera-based systems can only detect and report VRUs located directly next to the truck in a timely manner. The activity status of the right-hand direction indicator has no effect on system performance.

### 3.3. Dynamic Tests with VRU Moving in Parallel

In the dynamic tests with a commercial vehicle and VRU moving in parallel without visual cover, the Mercedes test vehicle showed a speed-dependent but not distance-dependent response behavior of the blind spot assistance system as you can see in Table 8.

If the truck overtook the VRU traveling at 10 km/h at a speed of 20 km/h, the information signal would be triggered as soon as the VRU entered the detection range of the blind spot assistance system from the front. If, on the other hand, the VRU overtook the truck at the opposite speed ratio, the information signal would be triggered when the VRU entered the detection zone from behind. If both test participants were moving at the same speed of 10 km/h, an irregular response behavior would be observed (see Figure 7). Shortly after the start, the information signal was triggered and deactivated again after a few meters. Only in one of the three runs did the signal subsequently remain for the rest of the way. If the test was run at a speed of 20 km/h instead, it would take longer for the signal to be activated, but the signal remained continuously until the end in all three runs.

When these tests were carried out with a visual obstacle between the right-hand side of the commercial vehicle and the VRU’s line of travel, the radar system was unable to detect the VRU at any time (Table 9).

The camera-based blind spot assistance system of the MAN test vehicle detected the VRU exclusively when it overtook the truck traveling at 10 km/h from behind at a speed of 20 km/h (Table 9, test number 3.2).

In this constellation, the information signal was output reliably, early, and constantly as soon as the VRU entered the detection range of the assistance system. In the other speed constellations tested, the VRU was not detected, and no information signal was induced.

After accident analysis, the following conclusions arise: Radar-based blind spot assistance systems can reliably detect a VRU driving in parallel at positive or negative differential speed. At approximately the same speed of truck and VRU, the higher the absolute speed level, the higher the detection reliability. In the case of camera-based systems, it is only possible to detect a VRU with sufficient certainty if it travels faster than the truck.

### 3.4. Dynamic Tests with Crossing Lines of Movement

In the dynamic tests with a simulated turning process of the commercial vehicle and a resulting crossing with the movement path of the VRU, the temporal response of information and warning signals to the vehicle driver was evaluated in particular as shown in Figure 8. In this setup, only the Mercedes test vehicle was tested in twelve runs. It was found that if the turn signal was not engaged, the information signal would be triggered first. If, on the other hand, the turn signal was activated, the warning signal would be triggered directly in four out of seven cases, and not the information signal first. Since the driver could not notice the visual signal if he did not look toward the A-pillar or into the instrument cluster at the moment of triggering, the following observations were based on the acoustic warning signal, which can be perceived regardless of the direction of gaze. The acoustic signal was triggered on average 0.1 s after the visual signal. First, the time was calculated for each run, which remained after the triggering of the acoustic warning signal until the theoretical collision. As shown in Figure 7, in cases (7)–(9), (16), (17), and (18), the turn signal was engaged on the right. In cases (7) and (9), it can be seen that they had a lower remaining time. In experiment number (7), the turn signal was not activated immediately at the start of the experiment, but only in the course of the approach. In the case of test number (9), the sequence and execution were observed correctly, so that here, a delayed detection by the assistance system can be assumed. Furthermore, it is noticeable that in the test runs (10)–(15) without the direction indicator switched on, the remaining time until the theoretical collision is on average more than four times shorter. The negative time indicated in test (14) means that here the warning signal did not occur until after the collision.

For the warning signal, these tests show a clear dependence on the activity state of the right-hand direction indicator. By activating it, the driver announces his intention to turn not only to the surroundings but also to the assistance system, so that the warning signal is triggered significantly earlier. The question whether the right turning indicator was activated or not will be even more important after an accident if the commercial vehicle was equipped with a blind spot assistance system. 

During the tests, as well as during the overhaul runs in public road traffic, it was found several times that the radar-based system of the Mercedes test vehicle triggered an information signal due to VRUs crossing behind the vehicle.

### 3.5. Integration of Blind Spot Assistance Systems into the Avoidability Analysis

In cases where the VRU was detected, the driver was warned, and yet an accident still occurred; thus, the question arises as to whether this leads to a significant change in the consideration of the driver’s reaction during the technical avoidability analysis.

In the following, our results are used as examples to show how a blind spot assistance system can be integrated into the technical avoidability analysis. The definition of an objective reaction prompt point for the commercial vehicle driver is difficult in the accident scenario since the VRU for the driver is only temporarily recognizable in the different exterior mirrors during the pre-collision phase. Due to the various critical areas to be observed during a right-turn maneuver with a heavy commercial vehicle in a confined urban environment, it is impossible to understand the driver’s gaze beyond doubt. Accordingly, no objective request for reaction can be determined. In the presence of a blind spot assistance system, on the other hand, the acoustic warning signal can be used as the latest possible reaction prompt point. In contrast to the purely visual information signal, this is perceptible to the driver regardless of the driver’s gaze.

The dynamic turning tests result in a considerable difference in the triggering times for the warning message depending on the driver’s timely actuation of the right-hand direction indicator during the pre-collision phase. If the direction indicator was activated early, the warning signal would be triggered 3.9 s to 4.9 s before the calculated theoretical collision point during four test runs (Figure 9). In one case, the acoustic warning was issued 1.9 s before the collision, as the direction indicator was activated late. In a test run with the direction indicator activated, the acoustic warning was triggered 1.7 s before the point of a collision without any obvious cause.

Without actuating the direction indicator, the blind spot assistance system is activated by the steering angle to the right, resulting in significantly later response times. In five test runs, the acoustic warning signal was triggered 0.15 s to 1.25 s before the calculated collision point. In one test, the warning appeared at 0.1 s after the collision (Figure 10).

The subsequent reaction time of the driver between the occurrence of the warning signal and the initiation of a defensive reaction in the form of a braking maneuver is highly dependent on the characteristics and abilities of the driver. It is subject to a corresponding range of fluctuation. In UN-ECE R151, a driver response time of 1.4 s is used to define the performance criteria of the blind spot assistance systems [5] and should, therefore, also be used in the present calculations. With the emergency braking deceleration of up to 7.36 m/s² measured in braking tests with the test vehicles, a braking time of 0.57 s can be calculated for a maximum permissible turning speed of 15 km/h.

Thus, it can be concluded from the tests that without the activation of the right-hand direction indicator, the acoustic warning message of the blind spot assistance system occurred too late for the driver to have been able to initiate a braking reaction in all the tests carried out compared to when a reaction time had been considered before the collision with the VRU.

When the direction indicator is activated, the acoustic warning is issued so that, in four out of six cases, a time window of 2.5 s to 3.5 s remained after the reaction time had elapsed to bring the vehicle to a standstill. In the two test runs with the delayed response of the acoustic warning, the available braking time was 0.3 s to 0.4 s. During this period, it would not have been possible to bring the vehicle to a complete standstill but to slow it down considerably, thereby significantly reducing the severity of the accident and, thus, the consequences of the accident.

Figure 11 shows the reconstruction result of the same traffic accident constellation without considering a blind spot assistance system. Here, it was investigated when the VRU would be recognizable to the commercial vehicle driver in the reconstructed approach constellation, in which indirect fields of view are presented via exterior mirrors or camera systems. The area visible in the exterior mirrors of the Mercedes test vehicle to the right of the vehicle was measured and drawn into a scaled plan view. Figure 11 shows that the VRU is typically perceptible in at least one indirect field of view over several seconds before the collision.

Accordingly, the accident would always be avoidable for the commercial vehicle driver when looking into the respective exterior mirror or the respective monitor at the right time. Since, as described, the driver’s gaze could be more comprehensible, from a purely technical point of view, neither an objective reaction prompt point can be defined, nor a meaningful avoidability analysis can be carried out in this accident constellation. According to the current state-of-the-art system, only a legal assessment of the avoidability for the commercial vehicle driver can be carried out here, considering the traffic situation. In the future, more attention will have to be paid to the activity status of the right-hand direction indicator and the time of activation. It must be kept in mind that the driver behaves in accordance with the rules when activating the direction indicator at an early stage.

## 4. Discussion

The knowledge gained has to be integrated into the common practice of traffic accident analysis with turning commercial vehicles and VRUs. To illustrate this, the following case study of a typical turning accident will serve as an illustration: At an inner-city intersection, a truck is in the front position at a red light. Since the truck wants to turn right at the intersection, the right-hand direction indicator is activated. Separated by a 2 m wide green strip, a cycle path runs parallel to the road, on which a cyclist approaches at a roughly constant speed of 12 km/h. When the truck starts after the traffic light changes to green, the cyclist is 18.5 m behind the cabin of the truck. Due to local conditions, the lateral distance between the right side of the truck and the cyclist’s driving line is 3 m. Since the truck driver does not notice the cyclist in the right wing mirror, he starts the turning maneuver at a speed of 13 km/h. At this point, the cyclist is 2.8 m to the right of the truck and 3 m behind the truck cabin. At unchanged speeds, the cyclist collides with the side of the truck immediately behind the front axle. Figure 12 shows the described situation.

If the truck is equipped with a radar-based blind spot assistance system that is fully functional, the accident could be easily avoided by the system as shown in Figure 13. About 7 s before the collision, the cyclist enters the detection range of the radar sensors from behind. Due to the truck’s starting process after switching traffic lights, the cyclist is faster than the truck at this time. From this point on, a visual information signal is issued by the blind spot assistance system. It is known from the tests carried out above that the visual and audible warning signal is activated no later than 3.9 s before the collision when the right-hand direction indicator is activated. Even if the driver only reacts to this latest possible warning signal, in this case study, he still has the reaction time of 1.4 s to initiate braking and stop the truck before crossing the driving line of the cyclist.

For the analysis of this typical accident scenario, first, the cause and the course of the accident need to be clarified. The blind spot assistance system investigated here is an exclusively warning function without any active intervention, although the question as to why the assistance system was insufficient to help avoid the accident might arise. Subsequently, an avoidability investigation must be carried out while also taking into account the blind spot assistance system.

### 4.1. Factors Influencing a Turning Accident Involving a Blind Spot Assistance System

After evaluating the tests carried out and considering the findings from the ADAC test series [1,10,19], the possible factors influencing the functional reliability of a blind spot assistance system can be systematically determined according to the current state-of-the-art system. As part of the reconstruction of a turning accident involving a commercial vehicle equipped with a blind spot assistance system, these possible sources of error must first be fully identified when the accident is recorded and systematically checked in the subsequent analysis to be able to reliably identify the cause of the accident.

In the first step, the commercial vehicle involved in the accident must be subjected to a technical vehicle inspection, whereby in the event of a turning accident, the focus should be on the visibility range from the driver’s seat and, if installed, on a blind spot assistance system.

The assistance system and its sensor and display components on or in the vehicle must be identified and subjected to a basic functional test. As explained above, basic conclusions can already be drawn from the type and installation of the environmental sensors about the detection range and detection reliability. The type and location of warnings issued to the driver determine the extent to which they can later be used as response prompt signals. In hardly any of the blind spot assistance systems examined was the internal monitoring of functionality carried out, and accordingly, no error message was sent to the driver in the event of a malfunction. An inspection of the assistance systems is still not part of the periodic technical inspection [23,24]. However, even if this were the case, a malfunction or total failure of the system due to an internal fault or, for example, sensor contamination that occurred immediately before the collision, could not be ruled out. This also applies to a possible system manipulation.

Like the fields of view of the mirror systems, the coverage area of the environment sensors should also be determined and measured. The results of the ADAC tests [1,10,19] showed that the activation of the blind spot assistance system, the coverage area size, or the activation of a supplementary camera monitor system is often linked to certain speed ranges that are also widespread. As we saw in our static tests, especially in a camera-based system, the detection range is not as wide as the area that can be seen in the camera–monitor system and even in longitudinal and lateral directions smaller than the range of a radar-based system. Depending on the situation at the accident site, there might be a great difference in the function of the assistance system according to the type of sensor that is used in the accident vehicle.

The ADAC found [1,10,19] that for systems with ultrasonic-based environment sensors, it can be assumed that any object on the right-hand side of the road will permanently trigger the blind spot assistance system since the ultrasonic system cannot differentiate between static objects and VRUs. The evaluation algorithms of camera-based systems can distinguish VRUs from static objects with high reliability so that hardly any false alarms are induced. In many systems, the classification of VRUs and the suppression of false triggering are based on the difference in speed to the commercial vehicle. In many cases, therefore, at a similar speed and a lower speed of the VRU compared to the commercial vehicle, no warning message is sent to the driver. A reliable warning is only triggered if the VRU overtakes the commercial vehicle from behind at a higher speed as our dynamic parallel driving tests show. In the case of radar systems, the detection range is highly dependent on the installation position of the vehicle. It is impossible to distinguish between static and dynamic objects with sufficient reliability in all systems as the ADAC found [1,10,19]. At the same time, radar-based systems can detect occasional false triggering by objects moving behind the vehicle at the edge of the detection area because of the wide detection range of this type of sensor.

As shown in the dynamic turning tests, the response of the blind spot assistance system is significantly dependent on the actuation state of the right-hand direction indicator. Clarifying this state of activity on the commercial vehicle involved in the accident is only possible through police witness interviews. In the future, it would be desirable if an event data recorder, which has yet to be introduced for heavy commercial vehicles, would also record and store the activity states of the lighting devices on the vehicle.

The warnings to the driver must be determined whether they are emitted exclusively visually or acoustically in the event of an acute risk of collision. In accident analysis, only an acoustic warning signal that is independent of the instantaneous direction of gaze can be used as an objective reaction prompt because it is not possible to find out where the driver was looking at the moment the warning was given.

The coverage area of the environment sensors determined in the vehicle inspection must be compared with the lateral distance between the movement corridors of the commercial vehicle and the VRU, which results from the course of road and the cycle path or footpath at the accident site. If the VRU is temporarily or constantly at the limit of the system’s specific detection range, it cannot be assumed that the blind spot assistance system will respond safely. Especially for camera-based systems, this lateral range is quite small.

If, during the pre-collision phase, there is a temporary or permanent obstruction of sight due to an obstacle between the vehicle and the VRU, this cannot be detected by the blind spot assistance system, and no warning message can be issued. Neither the systems tested by the ADAC nor the systems in our own experiments could at any time detect a VRU behind an obstacle. When assessing possible obstacles, attention must be paid to the position of the environment sensors on a commercial vehicle.

### 4.2. Integration of Blind Spot Assistance Systems into the Avoidability Analysis

To be able to understand the interaction between the warning blind spot assistance system and the driver during a turning accident, tests must be carried out, considering the relevant external and internal factors. In addition to the accident recording according to the current state of the art, the performance data of the blind spot assistance system must also be collected as described above. The course of the accident reconstructed based on these facts with unavoidable tolerance ranges can then be reconstructed, whereby the determined upper and lower limits of the tolerance ranges must be examined in more detail using experiments. As a rule, the VRU will have been seriously or fatally injured, so the primary decision must be made as to which limit values are to be applied in favor of the driver of the commercial vehicle involved in the accident.

If the assistance system is found to be functioning correctly, the acoustic warning signal of the assistance system can be understood as an objective request to react, since the driver must initiate braking at the latest in response to this warning of a concrete risk of accident. As described above, the triggering point depends on both the respective system and the environment, so that corresponding tests must be carried out. The activity status of the right-hand direction indicator plays a significant role here.

Depending on the remaining time between the occurrence of the warning signal and the theoretical collision point, the assistance system can have a positive or negative effect for the driver in the avoidability consideration. If the warning occurs in time, as in our experiments, the assistance system provides a clear avoidability of the accident for the driver of the commercial vehicle. However, in the case of larger distances between the VRU and the commercial vehicle or visual obscuration, constellations are also possible in which the warning occurs very late or not at all. In this case, the accident may no longer be avoidable for the driver when reacting to the warning signal.

## 5. Conclusions

The tests carried out showed that the blind spot assistance systems installed in commercial vehicles with an exclusive warning function can help to avoid an inner-city turning accident with a VRU moving in parallel. The technical characteristics of the environmental sensors used along with the accident location limit the reliability of the assistance systems, so they could not avoid every accident.

From a technical point of view, the acoustic warning signal of a blind spot assistance system can be used as an objective reaction prompt point for the commercial vehicle driver in a critical turning situation. The timing of the warning signal depends decisively on whether the driver activated the right-hand direction indicator in good time before the junction in the approach phase. With only an activated direction indicator, the full range of functions of the blind spot assistance system is available and can warn within a sufficient timespan prior to an imminent collision with a VRU.

The operating status of the direction indicator will not only play a role in the avoidability analysis for the VRU, but will also influence the perceptibility of the VRU by the driver in the avoidability analysis. On the one hand, the direction indicator shows the VRU the commercial vehicle’s intention to turn, so that the VRU can initiate an avoidance reaction if necessary. On the other hand, the activation of the direction indicator on the commercial vehicle also contributes to the activation of the blind spot assistance system, which supports the driver in detecting a VRU in time.

With the investigations and considerations presented here, it is now possible to integrate a blind spot assistance system into the technical–forensic accident reconstruction from cause research to avoidability analysis. After checking the correct function of the assistance system, the triggering time of the warning message can be determined under the respective circumstances of the accident, thus defining an objective reaction request point for the driver of the commercial vehicle. In this way, it will be easier to decide from a technical point of view whether a turning accident would have been avoidable for the driver or not.

For the moment, the legal assessment of the influence of the blind spot assistance system on the course of the accident and the avoidability analysis for the commercial vehicle driver remains unclear. Presently, assistance systems are considered as pure support so that the responsibility for driving behavior and, thus, an accident always remains with the driver without restriction. From a legal point of view, the avoidability study usually needs to consider the potential and the correct functioning of assistance systems installed in the vehicle. From the authors’ point of view, the assistance systems examined here as well as other assistance systems have the potential to influence the course and avoidability of an accident so significantly that a more differentiated legal view would be appropriate. Suppose the influence of the assistance system can be reconstructed with sufficient certainty. In that case, it is still possible to offer a much more far-reaching, objectively comprehensible decision-making aid through technical reconnaissance than was previously the case in the accident constellation under consideration.

The next technological step will be the introduction of blind spot assistance systems that actively intervene in the vehicle’s control system and initiate a braking maneuver when a dangerous situation is detected. Here, the reliability of the systems in detecting and classifying VRUs is even more important in order to avoid false trips. To ensure this, as well as a timely response, radar-based environment sensor systems should be used, as the tests have shown that they show the highest reliability. However, there is still room for improvement in the evaluation software of radar systems in order to be able to reliably detect a VRU moving almost as fast as the truck, especially in the low-speed range.

Furthermore, even with a precise reconstruction of the course of the accident and correspondingly detailed and reproducible tests about the triggering times of the blind spot assistance system, a tolerance range remains that could only be further reduced by the introduction of an event data recorder (EDR) for heavy commercial vehicles. In addition to driving data, this EDR should also contain information on warning and active interventions by driver assistance systems and the status of the turning indicators to be able to understand the influences of these systems reliably.

## Figures and Tables

**Figure 1 sensors-24-01517-f001:**
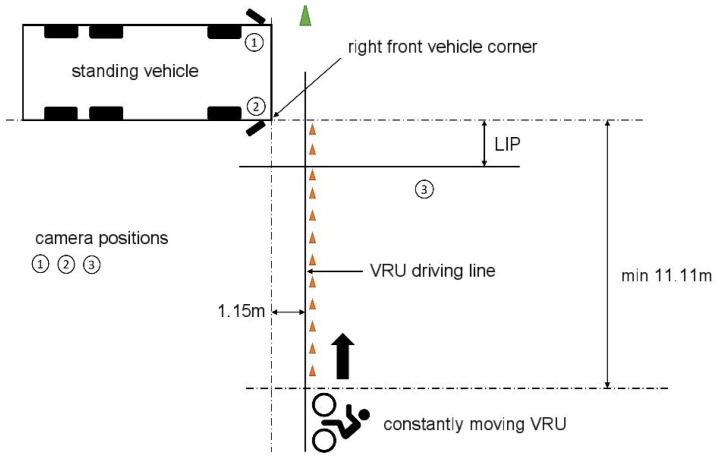
Static experimental setup with the VRU crossing the front of the commercial vehicle.

**Figure 2 sensors-24-01517-f002:**
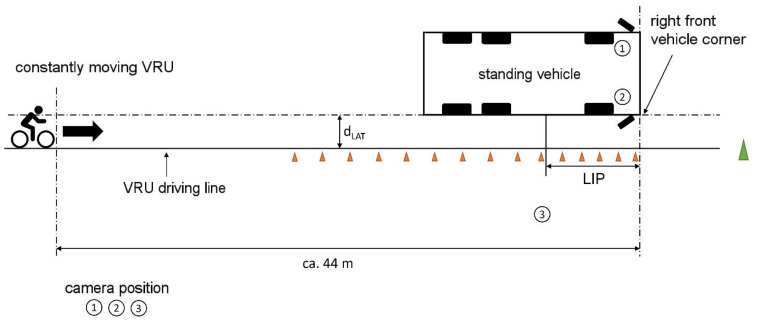
Static experimental setup with VRUs driving parallel to the commercial vehicle.

**Figure 3 sensors-24-01517-f003:**
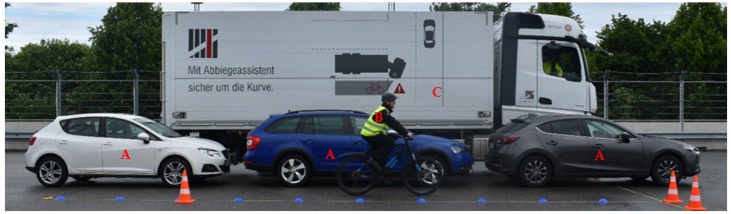
Static test setup with vehicles (A) as visual obstruction between the VRU (B) and the sensors of the commercial vehicle (C).

**Figure 4 sensors-24-01517-f004:**
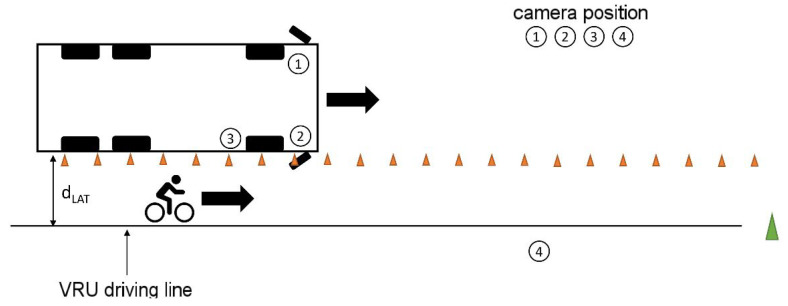
Dynamic test setup with parallel movement of commercial vehicle and VRU.

**Figure 5 sensors-24-01517-f005:**
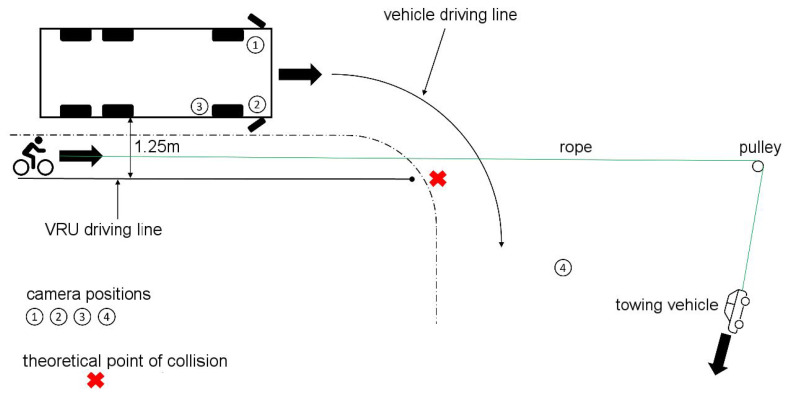
Dynamic experimental setup with the intersection of the movement corridors.

**Figure 6 sensors-24-01517-f006:**
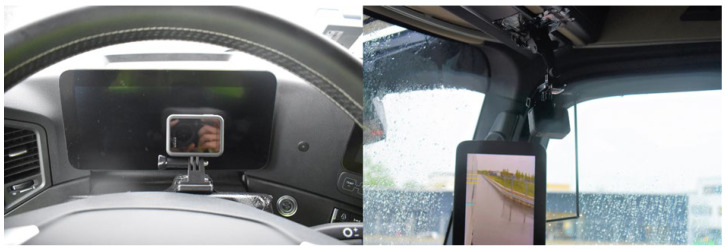
Mounting action cameras to record instrument cluster display and mirror fields of view.

**Figure 7 sensors-24-01517-f007:**
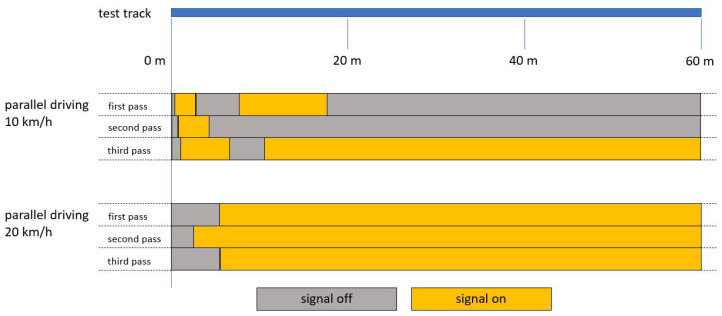
Time course of the information signal during tests in parallel travel at the same speed at a lateral distance of 1.25 m without visual obscuration.

**Figure 8 sensors-24-01517-f008:**
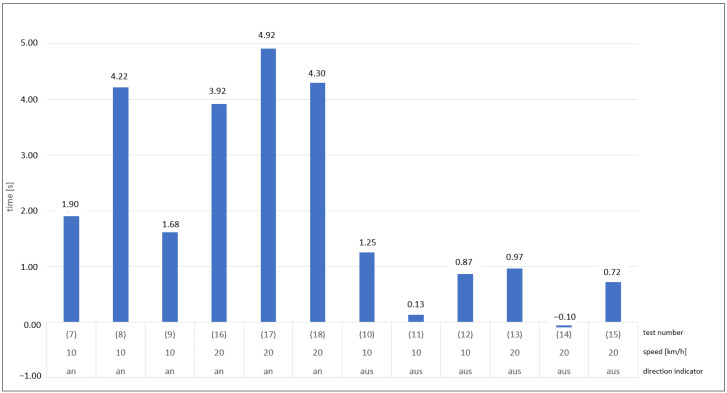
Remaining time between the occurrence of the acoustic warning signal and the theoretical collision.

**Figure 9 sensors-24-01517-f009:**
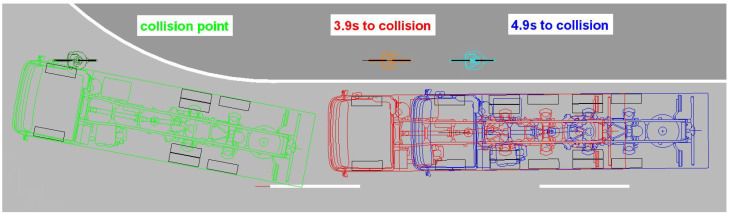
Shutter release range of the blind spot assistance system when the direction indicator is activated.

**Figure 10 sensors-24-01517-f010:**
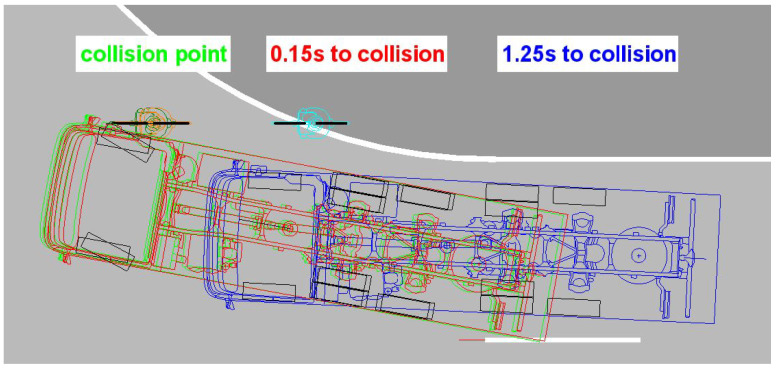
Blind spot assistance system triggering range without direction indicator activated.

**Figure 11 sensors-24-01517-f011:**
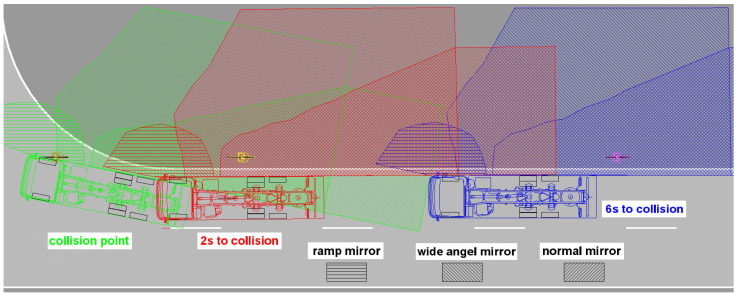
Perceptibility of the VRU in different indirect fields of view during the approach.

**Figure 12 sensors-24-01517-f012:**
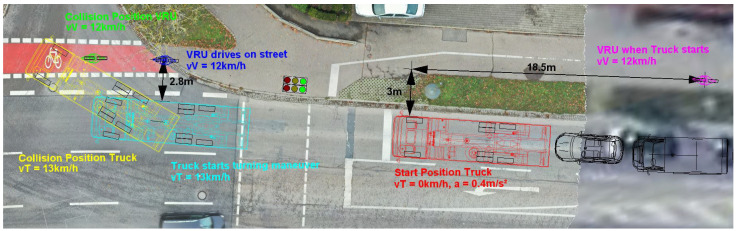
Typical accident between a VRU and a right-turning commercial vehicle.

**Figure 13 sensors-24-01517-f013:**
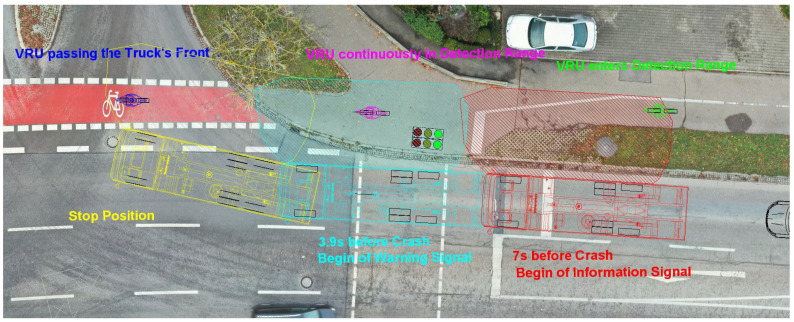
Situation with a perfectly working blind spot assistance system.

**Table 1 sensors-24-01517-t001:** Test matrix: VRU crossing the front of the standing commercial vehicle.

Test Number	Vehicle Speed [km/h]	VRU	VRU Speed [km/h]	LIP [m]
1.1	0	pedestrian	5	2
1.2	0	pedestrian with bicycle	5	2
1.3	0	cyclist	10	4
1.4	0	cyclist	15	6
1.5	0	cyclist	20	8

**Table 2 sensors-24-01517-t002:** Test matrix: static experiment with the VRU driving parallel to the commercial vehicle.

Test Number	Vehicle Speed [km/h]	VRU	VRU Speed [km/h]	d_LAT_ [m]	LIP [m]	Visual Obstruction
2.1	0	pedestrian	5	1.25	1.94	no
2.2	0	pedestrian with bicycle	5	1.25	1.94	no
2.3	0	cyclist	10	1.25	3.88	no
2.4	0	cyclist	15	1.25	5.82	no
2.5	0	cyclist	20	1.25	7.78	no
2.6	0	pedestrian	5	4.25	1.94	no
2.7	0	pedestrian with bicycle	5	4.25	1.94	no
2.8	0	cyclist	10	4.25	3.88	no
2.9	0	cyclist	15	4.25	5.82	no
2.10	0	cyclist	20	4.25	7.78	no
2.11	0	pedestrian	5	4.25	1.94	yes
2.12	0	pedestrian with bicyle	5	4.25	1.94	yes
2.13	0	cyclist	10	4.25	3.88	yes
2.14	0	cyclist	15	4.25	5.82	yes
2.15	0	cyclist	20	4.25	7.78	yes

**Table 3 sensors-24-01517-t003:** Test matrix: dynamic test with the parallel movement of commercial vehicle and VRU.

Test Number	Vehicle Speed [km/h]	VRU Speed [km/h]	d_LAT_ [m]	Visual Obstruction
3.1	10	10	1.25	no
3.2	10	20	1.25	no
3.3	20	20	1.25	no
3.4	20	10	1.25	no
3.5	10	10	4.25	no
3.6	10	20	4.25	no
3.7	20	20	4.25	no
3.8	20	10	4.25	no
3.9	10	10	4.25	no
3.10	10	20	4.25	no
3.11	20	20	4.25	no
3.12	20	10	4.25	no

**Table 4 sensors-24-01517-t004:** Test matrix: dynamic experimental setup with the intersection of the movement corridors.

Test Number	Vehicle Speed [km/h]	VRU Speed [km/h]	d_LAT_ [m]	Visual Obstruction	Direction Indicator
4.1	10	20	1.25	no	no
4.2	10	20	1.25	no	yes
4.3	20	20	1.25	no	no
4.4	20	20	1.25	no	yes

**Table 5 sensors-24-01517-t005:** Result matrix of experiment 1 with the distance between the VRU and Mercedes truck at the first information signal (“I” means that the right direction indicator was activated).

Test Number	LIP [m]	Distance at the First Information Signal [m]		
1.1	2	3.5	3 I	3.5
1.2	2	2.5	2.5 I	2.5
1.3	4	2	2.5 I	1.5
1.4	6	1	1.5 I	1
1.5	8	1.5	0.5 I	-

**Table 6 sensors-24-01517-t006:** Result matrix of experiment 2 with the distance between the VRU and Mercedes truck at the first information signal (“I” means that the right direction indicator was activated).

Test Number	LIP [m]	Visual Obstruction	Distance at the First Information Signal [m]		
2.1	1.94	no	10.5	10.5	10.5 I
2.2	1.94	No	10	9.5	10 I
2.3	3.88	no	9	8.5	9 I
2.4	5.82	no	7	7.5	8 I
2.5	7.78	no	9	9.5	5.5 I
2.6	1.94	no	10.5	10.5 I	10.5
2.7	1.94	no	9.5	9.5 I	10
2.8	3.88	no	9	9 I	9
2.9	5.82	no	8.5	8.5 I	9
2.10	7.78	no	8.5	8.5	8.5 I
2.11	1.94	Yes	-	- I	-
2.12	1.94	Yes	-	- I	-
2.13	3.88	Yes	-	- I	-
2.14	5.82	Yes	-	-	- I
2.15	7.75	Yes	-	-	- I

**Table 7 sensors-24-01517-t007:** Result matrix of experiment 2 with the distance between VRU and MAN truck at the first information signal (“I” means that the right direction indicator was activated).

Test Number	LIP [m]	Visual Obstruction	Distance at the First Information Signal [m]		
2.1	1.94	no	5	4.5 I	5
2.2	1.94	no	5.5	5.5 I	5
2.3	3.88	no	4.5	5 I	4.75
2.4	5.82	no	4.5	4	4.5 I
2.5	7.78	no	4	0.5	2 I
2.6	1.94	no	-	-	-
2.7	1.94	no	-	-	-
2.8	3.88	no	-	-	-
2.9	5.82	no	-	-	-
2.10	7.78	no	-	-	-

**Table 8 sensors-24-01517-t008:** Result matrix of experiment 3 with the activation status of the information signal of the Mercedes truck at different speed ratios.

Test Number	Vehicle Speed [km/h]	VRU Speed [km/h]	d_LAT_ [m]	Visual Obstruction	Information Signal		
3.1	10	10	1.25	no	partly	partly	Partly
3.2	10	20	1.25	no	yes	yes	Yes
3.3	20	20	1.25	no	partly	partly	partly
3.4	20	10	1.25	no	yes	yes	yes
3.5	10	10	4.25	no	partly	partly	partly
3.6	10	20	4.25	no	yes	yes	yes
3.7	20	20	4.25	no	partly	partly	partly
3.8	20	10	4.25	no	yes	yes	yes
3.9	10	10	4.25	yes	no	no	no
3.10	10	20	4.25	yes	no	no	no
3.11	20	20	4.25	yes	no	no	no
3.12	20	10	4.25	yes	no	no	no

**Table 9 sensors-24-01517-t009:** Result matrix of experiment 3 with the activation status of the information signal of the MAN truck at different speed ratios.

Test Number	Vehicle Speed [km/h]	VRU Speed [km/h]	d_LAT_ [m]	Information Signal		
3.1	10	10	1.25	no	no	no
3.2	10	20	1.25	yes	yes	yes
3.3	20	20	1.25	no	no	no
3.4	20	10	1.25	no	no	no

## Data Availability

The data presented in this study are available on request from the corresponding author.

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
