# Peer review of "Influence of Blind Spot Assistance Systems in Heavy Commercial Vehicles on Accident Reconstruction"

_sensors, 2024, doi:10.3390/s24051517_

Round 1
Reviewer 1 Report
Comments and Suggestions for Authors
The paper studies the blind spot assistance systems, which have certain practical significance. Tests with different systems and accident constellations were evaluated. The experimental design is proper. The results are clearly presented.
The reviewer has a question, how to ensure that the warning occurs in time.
Reviewer 2 Report
Comments and Suggestions for Authors
The paper provides many experiments results to evaluate the performance of blind spot assistance systems.
The design of the experiments are quite comprehensive.
However, the results should have been described in a more systematic and quantitive way.
The tests should have been repeated multiple times so that confidence intervals could be calculated.
Since there is not much technical depth in the experiments, the results should be presented in a more systematic way so that the results can be easily used for other purposes such as forensic accident analysis and avoidability analysis.
Comments on the Quality of English Language
English language of the paper is good.
Some editing might be needed. For example, the following sentence is not complelte.
"But although it might be of interest why the assistance system could not avoid the accident."
Reviewer 3 Report
Comments and Suggestions for Authors
Dear Author
Your manuscript provides a thorough examination of blind spot assist systems in commercial vehicles and their integral role in traffic accident analysis, a subject critical to the enhancement of road safety. This review aims to offer constructive insights to strengthen both the academic robustness and practical impact of your work. The blind spot assist system is undeniably crucial for improving road safety. I suggest emphasizing its implications in real-world applications, highlighting the benefits such technology could offer to policymakers and vehicle manufacturers. Additionally, your research findings have the potential to engage experts in the field deeply. To enrich the academic dialogue, it would be beneficial to articulate how your research provides novel insights that extend beyond current literature. To capture and maintain reader interest, consider including one or two case studies in the discussion section to demonstrate the system's effectiveness. It is also essential for the manuscript to detail experimental methodologies comprehensively, ensuring that peers can replicate your study accurately, which is fundamental to the ethos of scientific rigor and openness. Your scholarly and industrial contributions are commendable. In the conclusion, offering a perspective on future research directions and contemplating how your findings could inform subsequent technological innovations would add considerable value. Lastly, a careful refinement of the manuscript's language will aid in enhancing its overall clarity and readability.
I wish you the best of luck in your ongoing research pursuits.
Comments on the Quality of English Language
Grammar and Syntax: The document demonstrates precise grammar, with sentence structures that are coherent and clear. Tense usage and subject-verb agreement are consistently appropriate, aligning with the standards of scientific writing.
Clarity and Precision: The research's context, methodology, results, and conclusions are articulated with clarity, utilizing specialized terminology effectively. This renders the content unambiguous for professional readers.
Consistency: Terminology, illustrations, and the labeling of tables and figures are employed with uniformity throughout the paper, without any notable inconsistencies or sources of confusion.
Readability: The document is logically structured, with seamless transitions between paragraphs, which sustains the reader’s interest and facilitates comprehension.
Concision: The author conveys information succinctly, avoiding superfluous redundancy. The text is compact, yet it delivers a substantial amount of information.
Overall, the English language quality of this paper is in accordance with the expectations for journal publications. It exhibits the hallmarks of clear, precise, and professional scientific writing. The manuscript does not present any major English language issues that would necessitate significant revisions.
Round 2
Reviewer 2 Report
Comments and Suggestions for Authors
The authors modified the draft based on the reviewer's comment.
I hope that the authors should carefully read the paper again and correct any possible errors.
Comments on the Quality of English Language
In Table 5 and 7, there is no "I". Please change the caption of Table 5.
In Table 6, some column heads are now shown correctly.
There are still some typos.
- any of the test runsFrom an accident => any of the test runs. From an accident
- warning message.3.2 Static tests with parallel moving VRU => subsection should start from a new line.
- was equipped with a blind spot assistance system. => too much space between words.
Author Response
Dear respected Reviewer,
Thank you very much for your valuable and constructive feedback.
The whole paper has been checked and corrected. I hope we found all the mistakes.
In Tables 5 and 7 you can find the "I" in the results columns two and three. Because of this the captions have not been changed.
Table 6 has been corrected.
The typos have been corrected.
Best regards.